# The Genetic Architecture of the Etiology of Lower Extremity Peripheral Artery Disease: Current Knowledge and Future Challenges in the Era of Genomic Medicine

**DOI:** 10.3390/ijms231810481

**Published:** 2022-09-09

**Authors:** Lăcrămioara Ionela Butnariu, Eusebiu Vlad Gorduza, Laura Florea, Elena Țarcă, Ștefana Maria Moisă, Laura Mihaela Tradafir, Elena Cojocaru, Alina-Costina Luca, Laura Stătescu, Minerva Codruța Bădescu

**Affiliations:** 1Department of Medical Genetics, Faculty of Medicine, “Grigore T. Popa” University of Medicine and Pharmacy, 700115 Iași, Romania; 2Department of Nefrology–Internal Medicine, Faculty of Medicine, “Grigore T. Popa” University of Medicine and Pharmacy, 700115 Iași, Romania; 3Department of Surgery II—Pediatric Surgery, “Grigore T. Popa” University of Medicine and Pharmacy, 700115 Iaşi, Romania; 4Department of Pediatrics, Faculty of Medicine, “Grigore T. Popa” University of Medicine and Pharmacy, 700115 Iași, Romania; 5Department of Morphofunctional Sciences I, “Grigore T. Popa” University of Medicine and Pharmacy, 700115 Iaşi, Romania; 6Department of Mother and Child, Medicine-Pediatrics, “Grigore T. Popa” University of Medicine and Pharmacy, 700115 Iaşi, Romania; 7Medical III Department-Dermatology, “Grigore T. Popa” University of Medicine and Pharmacy, 700115 Iaşi, Romania; 8Screening Center for Oncological Diseases, Regional Institute of Oncology, 700115 Iași, Romania; 9Department of Internal Medicine, “Grigore T. Popa” University of Medicine and Pharmacy, 16 University Street, 700115 Iași, Romania; 10III Internal Medicine Clinic, “St. Spiridon” County Emergency Clinical Hospital, 1 Independence Boulevard, 700111 Iași, Romania

**Keywords:** lower extremity peripheral artery disease, atherosclerosis, genetic risk factors, heritability, polymorphism, modifier genes, epigenetics, GWAS, PRS

## Abstract

Lower extremity artery disease (LEAD), caused by atherosclerotic obstruction of the arteries of the lower limb extremities, has exhibited an increase in mortality and morbidity worldwide. The phenotypic variability of LEAD is correlated with its complex, multifactorial etiology. In addition to traditional risk factors, it has been shown that the interaction between genetic factors (epistasis) or between genes and the environment potentially have an independent role in the development and progression of LEAD. In recent years, progress has been made in identifying genetic variants associated with LEAD, by Genome-Wide Association Studies (GWAS), Whole Exome Sequencing (WES) studies, and epigenetic profiling. The aim of this review is to present the current knowledge about the genetic factors involved in the etiopathogenic mechanisms of LEAD, as well as possible directions for future research. We analyzed data from the literature, starting with candidate gene-based association studies, and then continuing with extensive association studies, such as GWAS and WES. The results of these studies showed that the genetic architecture of LEAD is extremely heterogeneous. In the future, the identification of new genetic factors will allow for the development of targeted molecular therapies, and the use of polygenic risk scores (PRS) to identify individuals at an increased risk of LEAD will allow for early prophylactic measures and personalized therapy to improve their prognosis.

## 1. Introduction

Lower extremity artery disease (LEAD), also called lower extremity peripheral artery disease (PAD) or lower extremity PAD, belongs to the large PAD family and is a chronic obstruction characterized by decreased blood flow to the arteries of the extremities of the lower limbs, which causes ischemia, with possible thromboembolic complications [1,2,3].

The most common stenosis is due to the development of an atherosclerotic plaque in the vascular wall, against the background of the presence of systemic atherosclerosis. For this reason, it has been suggested that LEAD could be considered a marker of widespread atherosclerosis and a predictor of future cardiovascular events [1,2,4].

In 2016, the American Heart Association (AHA) and the American College of Cardiology (ACC) updated information on the diagnosis and management of lower extremity PAD/LEAD [2,5]. In this guideline, the recommendations are limited to arterial atherosclerotic disease of the lower extremities (LEAD), and include disease of the aortoiliac, femoropopliteal, and infrapopliteal arterial segments [2,5,6]. In 2017, the European Society of Cardiology (ESC), together with the European Society for Vascular Surgery (ESVS), made an update of their own guidelines in 2011. Unlike American counterparts, Europeans have resorted to a broader definition of peripheral vascular disease (PVD) [2,6].

In the European document, the term “peripheral arterial diseases” includes all arterial diseases other than the coronary arteries and the aorta. The PADs family includes other peripheral localizations possibly affected by atherosclerosis, including the carotid and vertebral arteries, upper extremities, and mesenteric and renal arteries. The term “peripheral arterial diseases” should be clearly differentiated from the term “peripheral artery disease”, which is often used for lower extremity artery disease (LEAD) [2,6].

In our paper, we will use the term lower extremity artery disease (LEAD), also called lower extremity peripheral artery disease (PAD), to refer only to atherosclerotic obstruction of the arteries of the lower limbs.

LEAD affects > 200 million people worldwide, over 40 million of which reside in Europe [1], while in the United States, approximately 8 to 12 million adults aged ≥ 40 years are affected by LEAD [2]. The disease usually occurs after the age of 50, with an exponential increase after the age of 65, and the rate of affected people reaching about 20% by the age of 80 [1,7].

The total number of individuals with LEAD is growing, with an increase of 23% in the last decade due to the growth of the total population, global aging, and increased incidence of smoking and diabetes mellitus (DM). In developed countries, symptomatic LEAD is more common in men (although the difference is attenuated in the elderly), while in low- and middle-income countries, the prevalence is generally higher in women than in men [7]. In most studies, symptomatic LEAD cases account for 1:3 to 1:5 of all LEAD patients. Data on the incidence of LEAD in Europe are insufficient. In one Danish study, 10% of men aged 65 to 74 years were diagnosed with LEAD, of which a third showed symptoms of intermittent claudication (IC) [1,8]. The prevalence of LEAD in the Swedish population aged 60 to 90 years was 18%, and that of IC was 7% [9]. In a cohort that included 6880 German patients aged >65 years, the prevalence of LEAD (defined by an Ankle-brachial index (ABI) < 0.90) was 18%, with only 1 in 10 presenting with typical IC [10].

LEAD is associated with a high morbidity and mortality. LEAD-related mortality was initially attributed to cardiovascular complications of atherosclerotic coronary artery disease (CAD) and ischemic stroke. The increased risk of cardiovascular death was associated with advanced age, a lower baseline ABI value, and a prior history of amputation. Despite this, in over 40% of patients diagnosed with LEAD, the cause of death is not related to cardiovascular factors [11].

LEAD is defined by arterial stenosis (>50%) in the lower limbs, indicated by a value of the ankle-brachial index (ABI) < 0.9 (normal values of ABI: 0.9–1.4) and the absence of a peripheral pulse [12]. Both AHA/ACC and ESC/ESVS guidelines recommend resting ABI as the first-line study and toe-brachial index as a back-up in the setting of significant calcification artifact for the diagnosis of LEAD [1,2]. ABI is a useful, non-invasive tool for both LEAD diagnosis and surveillance and a powerful marker of generalized atherosclerosis and cardiovascular risk [1,2]. However, the diagnostic value of ABI for patients with LEAD is considerably limited in patients with calcified or non-compressible vessels (e.g., in patients with DM or with end-stage renal disease—ESRD), in whom false elevated ABIs may occur [13,14]. In these cases, additional imaging investigations (plethysmography method, duplex ultrasound, computed tomography angiography, magnetic resonance angiography) allow for the identification of the affected anatomical regions, especially in patients with indications for surgery or endovascular revascularization [13].

LEAD risk factors are classified into two categories: modifiable risk factors (hypercholesterolemia, smoking, DM (diabetes mellitus), systolic hypertension, sedentary lifestyle), and non-modifiable risk factors (age > 50 years, sex, family history of LEAD, chronic kidney disease) (Figure 1) [1,2,3,4,6,7,8,12,14,15].

In addition to traditional risk factors for LEAD, numerous studies have been conducted over time that have focused on identifying new possible mediators (biochemical markers) of LEAD [13,15,16]. Although some of the common risk factors for LEAD (e.g., dyslipidemia, diabetes, hypertension) themselves have genetic determinism, susceptibility to LEAD could be influenced by genetic modifiers (modifier genes), as well as a number of epigenetic changes which act independently of the traditional LEAD risk factors [17,18,19].

The identification of all these factors could shed light on the pathophysiological mechanism of LEAD and could allow for the development of new diagnostic methods and innovative therapeutic approaches [17,18].

In recent years, with the development of new molecular genetics technologies, major advances have been made in understanding the complex pathophysiology and deciphering the genetic architecture of multifactorial diseases, determined by the interaction between many genetic (polygeny) and environmental factors.

Although most studies have focused on atherosclerotic coronary artery ischemic disease (CAD), significant progress has also been made with regard to LEAD heritability. However, in contrast to CAD, relatively few susceptibility alleles for LEAD have been identified. This may be due to the clinical and genetic heterogeneity of LEAD [17,18].

In the last decade, progress has been made in identifying genetic variants associated with LEAD, at the level of the entire genome or exome, as well as in epigenetic profiling [17,20,21].

The identification of allelic risk variants that cause increased susceptibility to LEAD and the development of polygenic risk scores (PRS) is an area of interest that could play an important role in identifying people at a high risk for LEAD. In their case, the implementation of early prophylactic measures could improve prognosis and quality of life and create the foundations for therapeutic approaches based on precision medicine [17,22].

The aim of our paper is to provide an in-depth analysis of the data available in the literature on the role of genetic factors in the etiology of atherosclerotic LEAD. We aimed to review the current knowledge on the genetic basis of atherosclerotic LEAD, and then discussed the challenges and future directions to be followed in deciphering the etiology of LEAD.

We also focused on the complex interaction between genetic and environmental factors and the role of modifier genes (epistasis) and non-genetic modifiers (epigenetic regulation) that may influence susceptibility to LEAD. Thus, we performed the most comprehensive analysis of the methods (i.e., candidate gene-based association studies, family-based studies, linkage analysis, whole genome or exome analyses, mendelian randomization studies) used in the study of genetic factors involved in the etiology of LEAD and their results. We highlighted the advantages and limitations of each type of study, as well as the perspectives that this data offers for the implementation of effective prophylactic measures in order to avoid severe complications (which sometimes require limb amputation), and, last but not least, progress toward the development of new drugs and personalized therapy.

## 2. Literature Search Strategies and Data Collection

The data synthesized and presented in this review was obtained by examining the literature (PubMed, Google Scholar, MEDLINE, OMIM, MedGen databases) and using the following keywords: atherosclerosis, peripheral artery disease (PAD), lower extremity artery disease (LEAD), genetic risk factors, heritability, genetic polymorphism, modifier genes, epigenetic regulation, Mendelian randomization studies, micro RNAs, polygenic risk scores (PRSs), candidate gene-based association studies (CGS), linkage studies (LS), genetic linkage analysis (LA), genome-wide association studies (GWAS), or whole exome sequencing (WES) (Table 1).

## 3. Phenotypic Variability of Lower Extremity Peripheral Artery Disease

LEAD is diagnosed based on clinical symptoms, correlated with ABI values <0.9, sometimes requiring additional imaging investigations or plethysmography [1,2]. Patients with LEAD may have heterogeneous clinical manifestations, classified according to Fontaine and/or Rutherford classifications (Table 2) [1,2,66,67].

In the early stages, patients with LEAD are asymptomatic or may show symptoms of IC, manifested by pain and cramps of lower limb muscles, especially in the leg, which worsens with gait and decreases at rest. Generally, IC appears after a period without walking pain [1,2].

In advanced stages, the disease is manifested by chronic limb ischemia, or “critical limb-threatening ischemia” (CLTI), and common complications include ulcers and gangrene, with a high risk of limb amputation and death. Due to systemic atherosclerosis, LEAD is frequently associated with an increased risk of acute myocardial infarction (MI), ischemic stroke, and death [1,2]. The annual mortality is estimated at 1-2% in cases with IC, while in cases with CLTI, it is 20%; the lower extremity amputation rate is below 1% per year in cases with IC and 25–40% in those with CLTI [1,2].

Lower extremity amputation (LEA) is a significant burden on global health systems, causing a significant increase in morbidity and mortality. The annual mortality rate varies by country, age, sex, anatomical level of amputation, and associated comorbidities (those being higher in patients with diabetes mellitus), but globally it is estimated between 12% and 58% [68,69].

Some studies have shown a higher rate of lower limb loss in African Americans (AA) with LEAD compared to non-Hispanic White individuals [12,70].

Vitalis et al. [71] performed a meta-analysis of some studies that reported the prevalence of LEAD in both the general population and diabetic patients, and compared the prevalence of LEAD in different ethnic groups [71]. They identified a higher prevalence of LEAD in Black people (*p* < 0.001) and lower among Asians (*p* < 0.001), compared to Whites [71].

The ARIC study revealed that there are patterns specific to the race and sex of individuals with LEAD, which cause differences, both in terms of disease risk and clinical manifestations [72]. The progression of IC to CLTI cannot be accurately estimated, as it is known that only a portion of patients with IC progress to CLTI. This aspect raised the issue of the existence of genetic modifiers that could modify/influence the evolution of the disease, given that arterial stenosis at the level of the lower limbs is already installed [13].

AHA/ACC guidelines recommend screening to identify asymptomatic patients with LEAD, based on physical examination and ABI measurement, with an aim to develop an effective therapeutic intervention and to monitor the progression of the disease, avoiding severe and lethal complications of LEAD. Numerous studies have proven that aggressive medical treatment of risk factors decreases the mortality and morbidity in patients with LEAD. However, there is no consensus for the use of screening even in patients at a high risk for LEAD (smokers or diabetics), as there is no clear evidence that it leads to clinically important benefits, such as prevention of LEAD progression to severe forms (CLTI). In addition, there is no clear evidence to demonstrate that screening all patients with LEAD for asymptomatic atherosclerosis in other arterial beds would improve clinical prognosis and prevent acute “events” such as MI or ischemic stroke [6,13].

From the point of view of the pathophysiological mechanism of LEAD, it represents a distinct form of atherosclerotic vascular disease that differs from CAD and atherosclerotic cerebrovascular disease, through specific clinical manifestations. The formation and progression of atherosclerotic plaques, followed by rupture and instability in the coronary and cerebrovascular arteries, are the basis for acute complications such as MI, sudden cardiac death (SCD), and ischemic stroke [1,2,13,46]. For reasons still unknown, such acute “events” are not specific to LEAD, which is caused by the progressive narrowing of the arteries of the lower limbs and obstruction caused by atherosclerotic plaques [17,18].

One possible explanation would be that, most likely, the genetic factors that interact with the environment and the biochemical pathways through which they act contribute differently to the production of LEAD compared to CAD and atherosclerotic cerebrovascular disease [18].

## 4. Heritability of Atherosclerotic Lower Extremity Peripheral Artery Disease: Current Knowledge

Atherosclerotic LEAD is considered a complex disease with a multifactorial etiology, the determinism of which is impacted in variable proportions by the intervention of genetic and environmental factors [19,73]. Deciphering the way in which these factors interact, as well as the identification of modifying genetic factors (modifier genes) or regulators of gene expression (transcription factors or epigenetic factors) that can act independently of traditional risk factors, will allow for an understanding of the complex pathophysiological mechanisms of LEAD and facilitate therapeutic approaches. The importance of genetic factors in the etiology of LEAD has been demonstrated by numerous clinical and population studies [17,18,19].

The heritability of LEAD is estimated to be between 20–58%, according to family-based association studies, twin studies, or GWASs [17,18,74]. Heritability represents the amount of phenotypic (observable) variation in a population that is attributable to individual genetic differences [75].

Family aggregation has been reported in many studies, with a family history of LEAD proving to be associated with an increased risk of disease [17,22,76]. Valentine et al. [77] have provided evidence of a strong familial aggregation of vascular disease, showing that both premature CAD and occult lower extremity atherosclerosis (determined by duplex ultrasonography) are more common in the families of patients with premature LEAD (onset before age 50) [77].

The first studies that began to elucidate the role of genetic factors in LEAD were familial aggregation studies that included monozygotic and dizygotic twin studies, sibship studies, and studies of families in which there was a family history of LEAD [17,22,76,77]. Overall, these studies suggested that genetic factors have a relatively moderate role in the etiology of LEAD. In addition, it is not yet clear the exact proportion of genetic factors that determine susceptibility to LEAD and act independently of the already known atherosclerotic risk factors. This aspect will likely be able to be clarified with the analysis of more and more numerous patient cohorts, and with the expansion of exome or whole genome analysis [76].

To date, there have been three family studies that have reported estimates of LEAD heritability. In all three studies, based on the determination of ABI for diagnosis, as well as the quantification of the severity of LEAD, it was estimated that 21% of the interindividual variation of ABIs is due to hereditary factors [13,17,22,76].

In the first study of 184 pairs of white male twins examined in 1995–1997 (94 monozygotic and 90 dizygotic white, male twin pairs) from the National Heart, Lung and Blood Institute (NHLBI), 45–48% of the global variability of ABI could be attributed to additive genetic effects after adjustment for atherosclerosis risk factors [17,78]. In this study the concordance rates for low ABI values (ABI ≤ 0.9) were 33% among monozygotic (MZ) twins and 31% among dizygotic (DZ) twins, both being significantly higher than one would expect to occur by chance alone [17,78]. The authors identified a limitation of this study related to the small number of cases analyzed and the fact that estimating the effects of genetic factors may be influenced by selective mortality and loss to follow-up of individuals at a high risk for LEAD. It is also known that studies in twins overestimate the effects of genetic factors due to the presence of a greater number of common environmental factors compared to studies in non-twin patients [17,78].

The analysis of the 21 pairs of discordant twins for low ABI values found that there was an increased probability that the twins with LEAD would be a persistent smoker and more sedentary than their normal-control brothers [17,78]. Overall, this study provided relatively little evidence for a genetic basis of LEAD, but emphasized the importance of environmental factors in the etiology of LEAD [13,17,22].

A genome-wide linkage scan for ABI in 1310 African American (AA) and 796 non-Hispanic White (NHW) hypertensive non-twin sibs participating in the Genetic Epidemiology Network of Arteriopathy (GENOA) study reported a heritability of 35.7% in NHW (*p* < 0.001) before adjustment for atherosclerosis risk factors and 21.2% (*p* = 0.006) after adjustment for risk factors [17,22,76,79].

In the Framingham Offspring Study, LEAD heritability was investigated in a group of siblings, including 1097 men and 1189 women. The estimated value of ABI heritability in this study was 27% before adjustment for covariates and 21% after adjustment for covariates (both *p* values < 0.0001), which are values similar to those reported in the GENOA study [17,22,76,79].

Another important limitation of the previously mentioned studies is the fact that the vast majority of the subjects analyzed had ABI values at the limit or within the reference range (>0.9); in their case, the correlation between ABI and the degree of atherosclerosis in the arteries of the lower extremities was absent or very low [76].

The heritability studies of ABI available to date seem to reflect, in particular, the degree of influence of genetic factors in the reference range, which can be different from the real genetic influence in the case of LEAD. In addition, it is suggested that the true contribution of genetic factors to the etiology of LEAD is underestimated due to the non-inclusion in the studies of some of the severe cases of LEAD, as those individuals have the highest probability of inheriting the genetic susceptibility variants for LEAD [17,76].

Overall, family studies have definitely proven that genetic factors play a role in the etiology of LEAD; however, it is clear that genetic susceptibility cannot be attributed to a single gene, but to several genes with additive effects or to a combination of genes. However, the degree of genetic influence, independent of traditional risk factors for LEAD, remains to be accurately quantified [17,18,22,76].

To better understand the heritability of LEAD, several studies have looked at the role of family history as an independent risk factor for LEAD [77]. A study based on the Swedish Twin Registry is one of the most significant for patients with symptomatic claudication or critical limb ischemia. In this study, the proportion of phenotypic variance attributed to genetic factors among twins with LEAD was 58% (95% CI, 50% to 64%), and 42% (95% CI, 36% to 50%) for nonshared environmental factors [17,74]. There was a discrepancy regarding odds ratio (OR) for LEAD in people whose twin had LEAD compared to people whose twin did not have LEAD: 17.7 (95% CI 11.7 to 26.6) for MZ twins and 5.7 (95% CI 4.1 to 7.9) for DZ twins, which supports the role of genetic factors in the etiology of LEAD [17,74].

In the San Diego Population Study (SDPS), which included 2404 patients with LEAD of both sexes and different ethnicities, aged 29 to 91 years, the family history of LEAD (defined as any first degree relative with LEAD) was associated with a 1.83-fold higher OR of LEAD (95% CI (1.03, 3.26), *p* = 0.04), an association which was stronger for severe prevalent LEAD (OR 2.42, 95% CI (1.13, 5.23), *p* = 0.02) [80].

In another LEAD study that included 2296 patients with LEAD (diagnosed in the laboratories of Mayo Clinic, Rochester, Minnesota, from October 2006 through June 2012) and 4390 controls, family history for LEAD was present mostly in patients with LEAD compared to the control group, with an OR of 1.97 (95% CI 1.60 to 2.42). The association was stronger in younger subjects (age < 68 years; adjusted OR 2.46, 95% CI 1.79 to 3.38) than in older subjects (adjusted OR 1.61, 95% CI 1.22 to 2.12) [17,81].

Over time, attempts have been made to identify the genetic factors involved in the etiology of atherosclerotic LEAD through linkage analysis (LA) studies and association studies (CGS, GWAS and WES) [13,17,19,22].

### 4.1. Linkage Analysis Studies

Linkage analysis (LA) studies different areas of the genome, simultaneously analyzing several polymorphic DNA markers (microsatellites and single nucleotide polymorphisms—SNPs) that could be associated with the hereditary transmission of the disease in that family. LA aims to identify the genomic region where the gene is located, as well as the disease-causing allelic variants. The results are expressed in the logarithm of the odds (LOD). Positive LOD indicates that co-segregation of two genetic markers is more likely, and negative LOD favors this less likely probability. A LOD score of 3 or higher is generally understood to mean that two genes are located close to each other on the same chromosome [17,18,19]. The next step is the mapping of the genomic region in connection with the genetic marker−phenotype association. The marker with the strongest statistical association is considered to be closest to the disease-causing mutation [22].

Two linkage studies demonstrated the relationship between different loci and LEAD [17,18,19]. In the first study that included 116 families (a total of 272 Icelandic patients with LEAD diagnosed angiographically or a history of surgical revascularization for symptomatic LEAD), Gudmundsson et al. [23] identified a region located on chromosome 1p31 (between 100–110 cM) significantly associated (LOD = 3.93; *p* = 1.04 × 10^−5^) with LEAD, even after canceling the effects of hyperlipidemia, high blood pressure (BP) (hypertension), and DM [23]. Several candidate genes that intervene in different stages of the pathogenesis of atherosclerotic LEAD (e.g., hyperlipidemia, blood pressure regulation, vascular matrix regulation, inflammation, coagulation) have been discussed, but the genetic causal variants could not be identified [18,19].

Kullo et al. [24] demonstrated the association of ABI with 250 microsatellite markers located on chromosomes 1p, 6q, 7q, and 10p in 1310 African American (AA) patients and on chromosomes 3p and 3q in 796 non-Hispanic Whites (NHW), belonging to hypertensive siblings [24]. Quantitative trait locus (QTL) linkage analyses have identified several chromosomal regions that may have genes that influence ABI. However, the study could not demonstrate an obvious linkage with the LEAD phenotype. The conclusion of the study was that ABI is a modestly heritable trait in AA and NHW hypertensive sibships [24].

Although it does not require a specific candidate gene and completely scans the genome, LA studies have failed to identify the genomic regions associated with LEAD. Probably, this could be related both to the lack of large family pedigrees, as well as the limited power for detecting moderate effect size genetic variants. LA are designed to identify susceptibility loci that hold genes with a major effect size, rather than those with a small effect size [17]. For this reason, LA are successfully used to identify monogenic mutations (transmitted according to Mendel’s laws), but they are less used in the case of multifactorial, polygenic diseases, which have a complex etiology [19,20,21,22].

### 4.2. Association Studies

Association studies are an alternative method of studying polygenic inheritance and usually use the candidate gene approach (CGA). The known pathophysiological mechanisms of the disease and the genes that are supposed to intervene in different stages are considered, and the hypothesis of the association of these genes with the respective disease is analyzed. In the case of LEAD, genes involved in lipid metabolism, lipoproteins, hypertension, and DM were analyzed. However, the incomplete knowledge of the mechanisms of the pathogenic mechanisms of LEAD (the so-called missing heritability) represents a limitation of this type of approach. Advances in recent years in molecular genetics have allowed the expansion of analysis at the level of the entire genome (GWAS) or exome (WES), bringing information that sheds light in the dark corner of the etiology of complex, multifactorial diseases [17,18,19].

#### 4.2.1. Candidate Gene-Based Studies (CGS)

Candidate gene-based studies (CGS) are usually structured as case-control or cross-sectional studies. CGS investigate differences in the allele frequency of a specific known variant between individuals with the disease and those without the disease (controls) among unrelated individuals. CGS starts from the a priori hypothesis of the association between the gene of interest and the disease and allows a fine-grained mapping of a causal variant, demonstrating a greater power to detect low-modest effect size genetic variants [17,18,22].

Compared to CAD, few studies based on CGS analysis have been reported in LEAD [18]. In general, these studies considered the candidate genes involved in the pathogenic processes that cause atherosclerosis, including lipid metabolism, inflammation, platelet aggregation, vascular smooth muscle cell (VSMc) migration, hemostasis, coagulation, homocysteine metabolism, and angiotensin-converting enzyme (ACE) [17,18,22].

The Candidate Gene Association Resource (CARe) Consortium performed a meta-analysis of ≈50,000 SNPs across ∼2100 candidates to identify genetic variants for ABI and LEAD. The study included 21,547 European and 7267 African American (AA) patients. Although it failed to identify allelic variants associated with LEAD, the study identified a significant association with ABI for two polymorphic variants: rs2171209 of *SYTL3* gene (β = −0.007, *p* = 6.02 × 10^−7^) (correlated with lipoprotein a) and rs290481 of *TCF7L2* gene (β = −0.008, *p* = 7.01 × 10^−7^ (correlated with type 2 diabetes mellitus) (T2DM). Later GWASs confirmed the association of these genetic variants with LEAD [18,19,22,26]. The study failed to identify the allelic variants associated with LEAD, however, the most important association with LEAD was in the case of smoking patients who presented the single nucleotide polymorphism (SNP) rs3745274 located on *CYP2B6* (OR 1.24, *p* = 4.99 × 10^−5^) [18,26].

Kardia et al. [27] analyzed the association of 435 SNPs in 112 candidate genes in the case of 1046 non-Hispanic White (NHW) patients (hypertensive siblings). They identified significant associations with LEAD in the case of two SNPs (rs891512 and rs1808593) located on the *NOS3* gene. Three SNP−risk factor interactions were identified: the rs1042713 polymorphism of the *ADRB2* Gly16 gene interacting with lipoprotein (Lp)(a), and the rs828853 and rs1299142 polymorphisms of the *SLC4A5* gene interacting with diabetes mellitus (DM). Another 25 SNP−SNP interactions (located in 29 different genes) were identified [18,27]. The combination of SNPs with risk factors and the interaction between them explained 17.85% of the ABI variation in the analyzed group [27]. Other identified SNPs are located at the level of genes associated with triglyceride metabolism (*TGFB3* and *SLC22A3*), C-reactive protein (*ADD2* gene), fibrinogen (*ATP6B1*), homocysteine level (*SLC17A2* and *PKRAR2B*), and lipoprotein (Lp)(a) (*SLC22A3*) [18,27].

Many of the case-control CGS that indicate associations between SNPs and the pathogenic mechanisms of the disease had false positive results, the main limitations being determined by the low statistical power able to detect modest effect size genetic variants, as well as the testing of a limited number of genes. Thus, they do not allow for the identification of new genetic factors associated with LEAD [17,18].

#### 4.2.2. Genome-Wide Association Studies (GWASs)

Unlike linkage analysis (LA) studies, which allowed for the identification of large effect, low-frequency allele diseases with Mendelian patterns of inheritance, GWASs are now able to identify genes associated with polygenic, complex diseases, in which the interactions between genes and environmental factors also intervene [17]. GWASs examine the co-segregation of polymorphic genetic markers (SNPs) distributed throughout the genome in families affected by LEAD. A rare allelic variant present in 1% of the population is considered polymorphism. It is estimated that approximately 3,000,000 SNPs (one SNP in every 1000 base pairs) are present throughout the genome (3 billion base pairs) [17,18,19,46].

With the development of GWASs, the molecular mechanisms underlying the pathophysiological processes in many complex diseases were elucidated. In cardiovascular diseases (CVD), GWASs have identified numerous genetic factors associated with traditional risk factors (dyslipidemia, obesity, DM, and hypertension) [17,18,19,46].

The first GWASs analyzed the association between the 9p21 locus and increased susceptibility for CAD and LEAD [17,18,22,29,30]. Murabito et al. [28] performed the largest meta-analysis of 21 population-based cohorts (which included 41,692 individuals of European ancestry, including 3409 subjects with LEAD), investigating over 2.6 million imputed SNPs using the HapMap resource (International HapMap Project, “haplotype map”) [28]. The group identified a single SNP rs10757269 located on 9p21 near the *CDKN2B* gene had the strongest association with ABI (β = −0.006, *p* = 2.46 × 10^−8^); this association has been replicated in subsequent studies. The group identified other SNPs associated with ABI, but no variant reached the genome-wide significance threshold. These are located in two genes previously reported to be associated with LEAD, *DAB21P* rs13290547 (*p* = 3.6 × 10^−5^) and *CYBA* rs3794624) (*p* = 6.3 × 10^−5^), respectively, and *LDLR* rs1122608 (*p* = 0.0026) associated with CAD [28].

Helgadottir et al. [29] analyzed 2600 LEAD patients from Iceland, Italy, Sweden, and New Zealand and showed that the rs10757278 SNP located on the 9p21 locus is not only associated with CAD but also with LEAD (OR 1.14, *p* = 6.1 × 10^−5^), abdominal aortic aneurysm (AAA) (OR 1.2, *p* = 1.2 × 10^−12^), and intracranial aneurysm (OR 2.5, *p* = 1 × 10^−6^) [29].

In a similar study which analyzed the role of the 9p21 locus in LEAD, Cluett at al. [30] showed that the C allele of SNP rs1333049 is associated with an increased prevalence of LEAD (OR 1.29, *p* = 0.012) and with a lower mean ABI, in the case of 2630 White LEAD patients with an average age of 76.4 years [30]. This association was independent of the presence of previously diagnosed MI and atherosclerotic risk factors [30].

A GWAS study by Thorgeirsson et al. [31] analyzed the role of nicotine addiction and smoking in the pathogenesis of certain diseases (in populations of European descent) [31]. They identified a common polymorphic variant (SNP rs1051730) located in the nicotinic acetylcholine receptor *CHRNA5/A3/B4* gene cluster on chromosome 15q25.1 [25] as being associated with the number of cigarettes smoked (OR 1.4, *p* = 7 × 10^−15^) as well as with lung cancer and LEAD (OR 1.19, *p* = 1.4 × 10^−7^) [31].

Gretarsdottir et al. [32] analyzed 1292 patients with abdominal aortic aneurysm (AAA) and 30,503 controls and identified that the A allele of rs7025486 located on chromosome 9q33 could be associated with both AAA (OR 1.21, *p* = 4.6 × 10^−10^) and LEAD (OR 1.14, *p* = 3.9 × 10^−5^) [32].

Although it is not specific for LEAD, a case-control GWAS of 177 patients with thromboangiitis obliterans (TAO) identified three SNPs (rs376511 in the *IL17RC* gene, rs7632505 in the *SEMA5B* gene, and rs10178082 in the *RPA3* gene) that were significantly associated with TAO in the Uighur population [33]. TAO is a progressive, recurrent inflammation and thrombosis of the small and medium-sized arteries and veins of the hands and feet. Although the pathophysiological mechanisms underlying the occurrence of TAO and LEAD are different, the results of this study could contribute to the elucidation of the genetic architecture of LEAD. The *IL17RC* gene located on chromosome 3q25.3 is involved in the regulation of the inflammatory response [34], and the *SEMA5B* gene [35] located on chromosome 3q21.1 regulates axon growth during the development of the nervous system [25,33,35].

In a study that included 195 Japanese patients with LEAD, Koriyama et al. [36] identified the *OSBPL10* locus located on chromosome 3p23-p22.3 as being associated with LEAD [36]. OSBPL10 is a member of the OSBP family of intracellular lipid receptors [25]. Since dyslipidemia is a major factor in atherosclerotic LEAD, it is assumed that the *OSBPL10* gene intervenes by changing the homeostasis of plasma lipids (cholesterol), the same mechanism also being associated with CAD and ischemic stroke [36]. Because the association did not reach genome-wide significance, additional replication studies to verify the results in the Japanese population are needed [36]. Other newly identified loci had a modest association with LEAD: rs2554503 in the *CSMD1* gene (*p* = 5.7 × 10^−5^; OR = 1.32, 95% CI 1.15–1.51) or rs235243 in the *VSP13D* gene (*p* = 0.04; OR = 1.18, 95% CI 1.01–1.37) [36].

Matsukura et al. [37] analyzed 785 Japanese patients with LEAD with ABI < 0.9, 20,134 controls, and over >400,000 SNPs [37]. They identified two genome-wide significant SNPs for LEAD: rs9584669 in the *IPO5/RAP2A* gene (located on chromosome 13q32.2) [25] (OR 0.58, *p* = 6.8 × 10^−14^), rs6842241 in the *EDNRA* gene (located on chromosome 4q31.22–q31.23) [25] (*p* = 5.3 × 10^−9^) and an SNP that was nearly genome-wide significant: rs2074633 in *HDAC9* (located on chromosome 7p21.1) [25] (*p* = 8.4 × 10^−8^) [37].

The most significant association with LEAD was in the case of the *IPO5* gene. The *IPO5* gene encodes the IPO5 protein, a member of the importin beta family that plays a role in promoting the excretion of apolipoprotein A-1, a protein that plays an essential role in the regulation of HDL and, therefore, intervenes in the formation of atherosclerotic plaques in the vascular wall [17,25,37,38,39].

Recent GWASs are based on genomic data held electronically (electronic medical records—EMR; electronic health records—EHR) and the advantages of these types of studies lie in the possibility of analyzing a larger number of patients with LEAD [82]. The Electronic Medical Records and Genomics (eMERGE) Consortium is currently working to link EHR files with genomic data [82,83,84].

Kullo et al. [40] performed a GWAS study based on electronic data in which they analyzed 537,872 SNPs in 1641 LEAD patients and 1604 controls [40]. They identified an important association with LEAD (without genome-wide significance) of the rs653178 polymorphism in the *ATXN2-SH2B3* locus (located on chromosome 12q24.12) [25] (OR 1.22, *p* = 6.46 × 10^−7^) [40]. The SNP rs653178 has pleiotropic effects and has previously been associated with multiple phenotypes including MI, immunological disorders, and hematologic traits [40]. The SNP rs653178 is in linkage disequilibrium (LD) (r (2) = 0.99) with a missense SNP (rs3184504) in *SH2B3*, a gene involved in vascular homeostasis and immune inflammatory response mechanisms [40].

In 2019, Klarin et al. [85] performed the largest LEAD GWAS to date based on electronic health record (EHR) data [85]. The testing was done in the Million Veteran Program (MVP) and UK Biobank genetic biorepositories [85]. In this analysis, the authors tested ≈32 million DNA sequence variants associated with LEAD (31,307 cases and 211,753 controls). The authors identified 19 susceptibility loci for LEAD, 18 of which had not been previously reported. The 19 genetic variants implicated many known LEAD risk factors, suggesting pathophysiological mechanisms that include lipids, T2DM, smoking, hypertension, and thrombosis [85]. The 9p21 locus presented a genome-wide significant association in a GWAS based on ABI, demonstrating its utility in the diagnosis of LEAD [85]. The authors analyzed the differences regarding the contribution of the identified DNA sequences to the occurrence of peripheral, coronary, and cerebral atherosclerosis. They showed that 14 genetic risk variants for LEAD are also associated with CAD and 12 genetic variants are associated with large cerebral artery stroke (LAS) [85]. The effect of loci involved in lipid metabolism (*SORT1*, *LPA*, and *LDLR*) was diminished for LEAD, in the case of patients who simultaneously manifested CAD or LAS [85]. This suggests that part of the risk for LEAD may be determined by comorbidities or common pathogenic pathways for atherosclerosis in each of the vascular beds. In the same GWAS, the *COL4A1* locus (previously associated with CAD and small vessel disease of the brain) was found to be associated with LEAD and CAD, but not with LAS [85].

A multi-ancestry GWAS meta-analysis of 521,612 individuals (67,162 stroke cases and 454,450 controls) discovered 22 new stroke risk loci, bringing the total to 32. The *COL4A1* locus was associated with the occurrence of small cerebral artery stroke (*p* = 1.4 × 10^−4)^, suggesting that this variant may act differently in the cerebral vascular bed [59].

Four of the genome-wide significant variants (the loci *RP11-359M6.3*, *HLA-B*, *CHRNA3*, and *F5* p.R506Q) were exclusively associated with LEAD [59]. The *CHRNA3* locus is associated with nicotine dependence, and *F5* Leyden plays a role in thrombosis, suggesting that smoking and thrombosis may play a greater role in LEAD compared to atherosclerosis in other arterial territories [59].

Abrantes et al. [86] analyzed the association between a number of mitochondrial single nucleotide polymorphisms (mtSNPs) and haplogroups in 1652 LEAD cases and 1629 controls from the eMERGE LEAD GWAS and 1241 venous thromboembolism (VTE) cases and 1278 controls from the GENEVA GWAS of venous thrombosis, but they could not identify any significant association between these SNPs and LEAD or VTE [86]. The conclusion of the study was that, unlike vascular diseases, stroke, and DM, common mitochondrial variants individually or in combination do not play a major role in LEAD and VTE susceptibility [86].

In conclusion, LEAD GWASs have so far identified three SNPs (rs9584669 in *IPO5/RAP2A*, rs6842241 in *EDNRA*, rs10757269 in *CDKN2B*) with a genome-wide significant association (*p* < 5 × 10^−8^). Several SNPs showing a strong association (but without genome-wide significance) and one SNP (rs2074633 in *HDAC9* gene) with borderline significance were identified [17,36,37,87]. A “borderline” association was defined by *p* > 5 × 10^−8^ and *p* ≤ 1 × 10^−7^ [17,87]. The pathophysiological mechanisms by which these SNPs act have not yet been established. SNPs identified to date through GWASs have generally not been replicated in independent studies, with the exception of the 9p21 locus. Although there is clear evidence that the 9p21 locus is associated with increased susceptibility to LEAD and atherosclerotic vascular disease, in general, the precise mechanism by which this locus confers susceptibility to atherosclerosis-related diseases is not known [17].

#### 4.2.3. Sequencing-Based Association Analysis for LEAD

In a study by Safarova et al. [88] the targeted sequencing of 41 genomic regions previously identified by GWASs as being associated with CAD revealed several common variants/genes that are associated with LEAD, thus highlighting the basis of shared genetic susceptibility between CAD and LEAD [88]. The study included 1749 patients with LEAD and 1855 controls of European ancestry. The strongest association with LEAD was in the case of loci for the *LPL* and *SH2B3* genes, and other loci of interest were *ABO* and *ZEB2* (*p* < 4.5 × 10^−5^) [88]. Although this study did not identify genetic variants and new loci associated with LEAD, it paved the way for the future use of new analysis methods for LEAD, based on whole exome sequencing (WES) [88].

#### 4.2.4. Other Genetic Polymorphisms Associated with the Pathophysiological Mechanisms of Peripheral Arterial Disease (LEAD)

The progressive narrowing of the arteries of the lower limbs, determined by atherosclerosis plaques, results in the appearance of the characteristic phenotypic manifestations of LEAD. Due to the progressive evolution of the disease, without acute events such as MI or ischemic stroke, it is suggested that the pathophysiological mechanism of LEAD, determined by the interaction between multiple genetic and environmental factors, is different from that which produces atherosclerosis in other vascular beds (CAD or atherosclerotic cerebrovascular disease) [13,19]. Atherosclerosis is a silent chronic vascular pathology, which causes most cases of ischemia of the lower limbs. The evolution of vascular disease involves extensive intimal lipid deposition, endothelial dysfunction, leukocyte recruitment, foam cell formation, exacerbated innate immune responses, VSMc proliferation, and extracellular matrix remodeling, leading to atherosclerotic plaque formation [89].

The genetic factors that cause LEAD are involved both in the initiation and formation of atherosclerotic plaques (genes involved in lipid metabolism and blood pressure regulation), as well as in plaque progression (genes involved in inflammation, thrombosis, platelet aggregation, vascular endothelial factors) (Figure 2) [13,17,22].
a.Polymorphism of Genes Involved in Lipid Metabolism

In a study that included 205 patients with LEAD, Monsalve et al. [41] showed that the R2 and X1 alleles of the *APOB* gene, located on chromosome 2p24.1 [25], which encodes apolipotrotein B (ApoB), were more prevalent in patients with LEAD compared to the control group [41]. The conclusion was that the polymorphism of the *APOB* locus causes an increased susceptibility to the development of arterial disease, without being able to specify exactly the place at the level of the vascular bed where the disease will develop [41].

In a prospective cohort study, the Secondary Manifestations of ARTerial disease (SMART) study, of 7418 patients with CVD including LEAD, Koopal et al. [42] analyzed the association between the *APOE* gene (located on chromosome 19q13.32) [25] promoter polymorphism and the occurrence of LEAD [42]. Starting from the known fact that certain *APOE* allelic variants are associated with CAD and stroke, they showed that, of the six known *APOE* genotypes (*APOE2/2*, *APOE2/3, APOE2/4, APOE3/3, APOE3/4, APOE4/4*), the homozygous *APOE2/2* (ε2/ε2) genotype is associated with an increased risk of LEAD in patients at a high risk of CVD. In these individuals, no association was observed between *APOE* genotypes with CAD, stroke, or vascular mortality [42].
b.Polymorphism of Genes Involved in the Mechanism of Inflammation in the Vascular Wall

Inflammation plays a major role in the pathogenesis of the development and progression of the atherosclerotic plaque, involving numerous molecules (cytokines, chemokines, adhesion molecules, and proteolytic enzymes) [13,19]. SNPs in genes encoding inflammatory molecules may determine genetic profiles associated with an increased risk of the development and progression of CVD [13,19].

The *IL6* gene (located on chromosome 7p15.3) [25] encodes interleukin 6 (IL6), a cytokine that regulates the production of C-reactive protein (CRP), an inflammatory marker associated with an increased risk of atherosclerotic CAD [25]. IL6 is produced by various types of cells, including monocytes, adipocytes, and endothelial cells. High levels of IL6 have been found in patients with atherosclerotic disease [13]. In a meta-analysis that included 50 studies (involving 33,514 subjects), Yin et al. [43] showed that the *IL6* promoter polymorphisms (−572G > C and −174G > C) influence *IL6* gene transcription and are associated with an increased level of circulating inflammatory markers and an increased risk for atherosclerosis [43].

Flex et al. [44] evaluated the association between peripheral artery occlusive disease (PAOD) and the *IL6* promoter polymorphisms in 84 patients with LEAD and 183 controls. They showed that the GG genotype and the G allele were significantly more common in patients with PAOD. These data support the hypothesis that the *IL6* G/C polymorphism is important in the pathophysiology and evolution of ischemic disease of the lower limbs, suggesting that IL6 plays a role in the pathogenesis of PAOD [44].

In another study of 157 LEAD patients and 206 controls, Flex et al. [45] analyzed pro-inflammatory genetic profiles in subjects with PAOD and CLTI [45]. Polymorphisms of the interleukin-6 (*IL-6*)-174 G/C, C-reactive protein (*CRP*) 1059 G/C, monocyte chemoattractant protein (*MCP-1*)—2518 A/G, macrophage migration inhibitory factor (*MIF*)—173 G/C, E-selectin (*SELEl*) Ser128Arg, matrix metalloproteinase (*MMP-1*—1607 1G/2G, *MMP-3*—1171 5A/6A, and *MMP-9*—1563 C/T), and intercellular adhesion molecule-1 (*ICAM-1*) 469 E/K were analyzed [45].

Polymorphisms of the genes *IL6*, *SELE* (located on chromosome 1q24.2, encodes selectin-E), *ICAM1* (located on chromosome 19p13.2, encodes intercellular adhesion molecule 1, ICAM1), *MCP1/CCL2* (located on chromosome 17q12, encodes monocyte chemoattractant protein-1, MCP-1), *MMP1* (encodes matrix metalloproteinases1, MMP1), and *MMP3* (encodes matrix metalloproteinase-3, MMP3)−both located on chromosome 11q22.2 [25]−were significantly and independently associated with PAOD [45]. In addition, the risk for PAOD and CLTI correlates with the number of risk genotypes present in the same individual, which is an example of how susceptibility to a disease results from the interaction between a series of modifier genes [45]. Considering this aspect, the authors concluded that the synergistic effects between proinflammatory genotypes could be potential markers for the presence and severity of atherosclerotic disease [13,45].
c.Genes Involved in Blood Pressure Regulation

##### Angiotensin-Converting Enzyme (ACE), Angiotensin II Type I Receptor (AGTR1), and Angiotensinogen (AGT) Genes Polymorphism

The *ACE* gene (located on chromosome 17q23.3) [25] encodes the angiotensin-converting enzyme (ACE) that converts angiotensin I to angiotensin II, a potent hypertensive factor [25]. The effect of angiotensin II is mediated by the angiotensin II type 1 receptor (AGT1R), encoded by the *AGTR1* gene (located on chromosome 3q23) [25]. ACE and angiotensinogen (AGT) (encoded by the *AGT* gene located on chromosome 1q42.2) [25], play an important role in regulating blood pressure, while AGTR1 plays a major role in the etiology of many CVD, including CAD [25,46].

The role of the renin−angiotensin system (RAS) in the pathogenesis of atherosclerotic CVD is evidenced by the clinical efficacy of ACE inhibitors in the treatment of a wide range of CVD [47]. There is also evidence to support the vascular protective role of ACE inhibitors and their role in the management of patients at cardiovascular risk (due to the cardioprotective effect) or with clinically manifested LEAD [13,47]. The *ACE* I/D polymorphism (rs4646994) is characterized by the presence (I) or absence (D) of a 287 bp Alu repeat sequence in intron 16, resulting in three genotypes: DD, II, and ID. The *ACE* D allele is associated with increased serum levels of circulating ACE enzyme [47,48].

There are several studies investigating the association between the *ACE* (I/D) polymorphism and LEAD, but the results remain controversial [49,50]. In a meta-analysis by Han et al. [48] which included 13 studies (1966 patients with LEAD and 6129 controls), the authors showed that *ACE* I/D polymorphism could be associated with an increased susceptibility to LEAD in the Asian population, but there is no clear evidence whether this is also true of Caucasians [48].

Bașar et al. [49] found a significant association between *ACE* I/D polymorphism and LEAD [49]. In their study, the serum level of the ACE enzyme was significantly lower in the group of patients with the homozygous genotype *ACE II* compared to those with the heterozygous genotype *ACE* ID or homozygous genotype DD (*p* < 0.05). Although conflicting results have been reported, the authors believe that the *ACE* ID genotype may be a risk factor for LEAD [49].

Fatini et al. [Fatini-50] analyzed the presence of the *ACE* I/D and −240A>T, *AGT* M235T, and *AGTR1* 1166 A>C polymorphisms in 281 LEAD patients and 485 controls. The authors emphasized the role of the *ACE* D/-240T haplotype in predisposing to LEAD, also in the absence of other atherosclerotic comorbidities. No influence of the analyzed polymorphisms on the severity of the disease (according to Rutherford categories) was found [50]. The results of Fatini et al. [50], were consistent with those reported in other studies and indicated the existence of an effect of the *ACE* gene regardless of the vascular bed where the atherosclerotic process is located, while in the case of the *AGT* and *AGTR1* genes, the effect is proven in the case of coronary localization, not in the case of extracoronary vascular beds [50].
d.Genes Involved in the Function of Vascular Smooth Muscle Cells (VSMc)

Vascular smooth muscle cells (VSMc) can be involved in the formation of the atherosclerotic plaque by at least two mechanisms: modulating blood pressure through vascular tone and vascular remodeling, which, together with other factors, lead to either stabilization or progression and rupture of the atherosclerotic plaque [19,46].

##### Endothelial Cell Nitric Oxide Synthase 3 (NOS3) Gene Polymorphism

ACE can also affect the degradation of bradykinin and the release of nitric oxide (NO) [13]. NO is synthesized from L-arginine under the action of the nitric oxide endothelial cell synthase (eNOS), an enzyme encoded by the *NOS3* gene located on chromosome 7q36.1 [25] and expressed in vascular endothelial cells, platelets, and cardiomyocytes [25]. Three isoforms of eNOS are synthesized, among which, those synthesized at the endothelial level are relevant for vascular disease. eNOS causes the release of nitric oxide (NO) at the vascular level, which causes the relaxation of vascular smooth cells and the adhesion and aggregation of platelets. NO also has the physiological role of preventing the formation of atherosclerotic plaques by inhibiting the proliferation of smooth muscles [25,46,51].

Ismael et al. [51] showed in a study that included 35 patients with IC, 26 patients with CLTI, and 35 non-LEAD controls, that the NO system and its regulators are significantly compromised in LEAD. This dysregulation appears to be driven by increased oxidative stress and worsens as the disease progresses from IC to CLTI [51]. Numerous studies have shown that certain *NOS3* gene polymorphisms cause a decrease in NO synthesis at the vascular level, causing endothelial dysfunction, which can play a role in the occurrence of CAD or hypertension [52,53].

The *NOS3*-786T (C allele) and NOS3 4a/4b polymorphisms are associated with low plasma levels of NO [25,53]. In the Edinburgh Artery Study, Fowkes et al. [54] showed that the presence of the A allele in intron 4 of the *NOS3* gene (*NOS3* 4a/4b polymorphism) was associated with an increased risk of CAD in non-smokers, but the *MTHFR* and *NOS3* genotypes had little influence on the risks of LEAD and CAD in the elderly population [54].

In a study by Sticchi et al. [55] which included 281 patients with LEAD and 562 controls, the presence of the *NOS3* -786C allele was significantly higher in patients with LEAD compared to the control group; this aspect was not found in the case of −894T and 4a alleles [55]. Both in the case of smokers and non-smokers, the simultaneous presence of *NOS3* -786C and *NOS3* 4a alleles had a significant association with the susceptibility to LEAD [55]. The authors demonstrated that the association between the *NOS3* haplotype -786C/4a and the *ACE* D allele increases the susceptibility to LEAD in smokers (but not in non-smokers), thus providing evidence regarding the interaction between modifier genes and environmental factors [55].
e.Genes Involved in Vascular Homeostasis

The *SLC2A10* gene (located on chromosome 20q13.12) encodes SLC2A10 (glucose transporter 10; GLUT10), a member of the facilitative glucose transporter family which plays a significant role in maintaining glucose homeostasis [25]. Loss-of-function mutations in the *SLC2A10* gene cause arterial tortuosity syndrome via upregulation of the TGF-β pathway in the arterial wall. The same mechanism could be the basis of the vascular changes in type 2 diabetes mellitus (T2DM) [13,25].

In the study by Jiang et al. [56], the common allelic variants of the *SLC2A10* gene (the T allele at rs2179357) showed the strongest association with LEAD in T2DM patients (*p* = 2.6 × 10^−4^; OR: 3.87; 95% CI: 1.97–7.58) [56]. The SNP rs2235491 (Ala260Thr) located in exon 2 of the *SLC2A10* gene was in strong linkage disequilibrium (LD) with rs2179357 (D′ = 0.82) but did not show a statistically significant association with LEAD [56]. Previous studies provided evidence that SNP rs2235491 Ala/Ala carriers presented a higher fasting plasma insulin level, and thus, this polymorphism could have contributed to the occurrence of vascular complications in patients with T2DM via associated hyperinsulinemia [56]. The involvement of the *SLC2A10* polymorphism in the occurrence of other micro or macrovascular complications is unclear, requiring further investigations [56]. The authors identified a common haplotype (H4) that confers an increased risk for LEAD in patients with T2DM carriers (*p* = 0.03, OR: 14.5; 95% CI: 1.3–160) compared to non-carriers [56].
f.Gene Polymorphisms Contributing to Atherothrombosis in LEAD

Thrombosis plays a major role in the progression of atherosclerotic plaques, which can be located at any level of the arterial vascular bed (coronary artery, aorta, carotid, and peripheral arteries). Plaque development and its rupture are distinctive signs of atherosclerotic CAD [19,89,90].

In recent years, a series of genetic polymorphisms involved in thrombophilia, a hereditary condition associated with an increased risk of venous thromboembolic events (VTE) have been identified [57]. It is not exactly known if these polymorphisms are also accompanied by an increased risk of arterial thrombosis, which causes ischemic stroke, MI, or PAOD, or their recurrence [57].

Prothrombin (FII) (encoded by the *F2* gene, located on chromosome 11p11.2) [25] has a role in fibrin formation, endothelial activation of platelet aggregation, and leukocyte recruitment [25]. The *F2* G20210A allelic variant causes increased plasma levels of prothrombin, associated with an increased risk of venous thrombosis. However, the role of this polymorphism in causing arterial thrombosis, as in the case of the Arg506G allele of the *F5* gene (encoding factor V Leiden—F5 Leiden, located on chromosome 1q24.2) [25], remains controversial [57].

Reny et al. [58] analyzed the association of *F2* and *F5* polymorphisms in 144 patients (with prior ischemic stroke) and LEAD [58]. They reported a statistically significant association between the *F2* G20210A allele, which is related to increased local thrombin production, and LEAD. The *F5* Arg506Gln polymorphism was not associated with a susceptibility to LEAD [58].

Human platelets express two platelet receptors for adenosine diphosphate (ADP): P2Y1 (coupled with Gq) and P2Y12 (coupled with G1). The P2Y12 receptor (encoded by the P2Y12 receptor gene, *P2YR12*, located on chromosome 3q21.5) [25], plays a central role in platelet aggregation at the site of vascular injury, both in physiological hemostasis and pathological thrombosis [60]. *P2YR12* gene polymorphisms have been shown to be correlated, not only with CAD, but also with LEAD. In the study by Fontana et al. [61] that included 184 patients with LEAD, there was an association between the *P2YR12* H2 haplotype and the alteration of platelet aggregation induced by ADP [61]. Thus, this polymorphism could be associated with an increased risk of LEAD, even after adjusting for traditional LEAD risk factors [61].

Fibrinogen β gene (*FGB*, located on chromosome 4q31.3) polymorphisms [25], have been shown to affect plasma fibrinogen levels increasing the risk for LEAD, stroke, and CAD [62]. In their study, Beahgue et al. [62] showed that the plasma fibrinogen level was significantly associated with β *Bcl I* (*p* < 0.015), β *C448* (*p* < 0.004), β *Hae III* (*p* < 0.002) and β-*1420* (*p* < 0.003) genetic variants and the increase in the plasma fibrinogen values was proportional to the number of less frequent alleles (codominant effect). Only two polymorphisms, β *Hae III* (*p* < 0.0003) and β-*854* (*p* < 0.01), were independently associated with increased plasma fibrinogen values, which correlate to an increased risk for MI in patients with severe CAD, especially in smokers [62].

Fowkes et al. [63] conducted a study that included 121 patients with LEAD and 126 controls, aged 55 to 74 years included in the Edinburgh Artery Study [63]. They showed that polymorphism of the fibrinogen β gene (*FBG*) (extended haplotypes of 4.2 kb for heterozygotes) was associated with an increased risk of peripheral atherosclerosis (LEAD). This influence is not only mediated by increased plasma values of fibrinogen, but could also be due to a structural variant of fibrinogen or a linkage disequilibrium (LD) with a neighboring gene [63].
g.Hyperhomocysteinemia

The enzyme methylenetetrahydrofolate reductase (MTHFR) encoded by the *MTHFR* gene (located on chromosome 1q36.22) is involved in the remethylation of homocysteine to methionine [25]. Mutations of the *MTHFR* gene, which cause a decrease in enzyme activity, are associated with increased plasma homocysteine levels, which are considered a risk factor for vascular diseases, proven in many studies [64].

The association between the *MTHFR* polymorphism and LEAD has been studied in the literature, but the results have been controversial. The Linz Peripheral Arterial Disease (LIPAD) study did not correlate the *MTHFR* polymorphism and LEAD [91], while, in a meta-analysis of 9 studies, Khandanpour et al. [65] showed that the homozygous genotype for the C677T allele was associated with an increased risk of LEAD (OR 1.36, 95% CI 1.09, 1.68) [65].

## 5. Discussions and Future Challenges

Deciphering the complex pathophysiological mechanisms and identifying all the genetic factors involved in the production of atherosclerotic LEAD is an objective of research in modern medicine. The translation of new information into medical practice through new generation molecular technologies is essential for the development of screening methods based on polygenic risk scores (PRSs). This new approach will facilitate the identification of people at high risk for LEAD, the implementation of early prophylactic measures, and the setting of new therapeutic targets. Gene therapy strategies are the next step in the treatment and prevention of the disease.

Despite improved treatment and prophylactic methods for atherosclerosis, the morbidity and mortality of LEAD remains elevated. Although most studies have focused on coronary atherosclerosis and CAD, significant progress has been made in recent years regarding the heritability of LEAD. The number of susceptibility loci for LEAD that were identified was much lower in LEAD compared to atherosclerotic CAD. This could be due to the clinical and genetic heterogeneity of LEAD [17,18].

In the last decade, extended analyses at the level of the entire genome or exome, as well as epigenetic profiling, allowed for the identification of new genetic variants associated with LEAD [17,20,21].

### 5.1. Challenges for the Future in the Post-GWAS Era

The main challenges in investigating the genetic basis of LEAD are represented by phenotypic variability correlated with genetic heterogeneity; the identification of modest effect size genetic variants; the identification of interactions between different genes (epistasis) and between genes and environmental factors; the identification of rare allelic variants that influence the susceptibility to LEAD; and the investigation of structural genetic variants associated with complex, polygenic diseases, including LEAD [17,18,22].

An important aspect in deciphering the genetic architecture of LEAD is represented by the need to analyze increasingly large cohorts of patients. In this sense, both the collaboration of international consortia, as well as the use of the information provided by electronic medical records (EMR) or the electronic health records (EHR) is necessary [17,18,22].

The creation of large, accessible international databases and the sharing of information between different study groups will elucidate the conflicting results of studies that included small groups of patients and will allow for meta-analyses on LEAD similar to those for CAD [17,18,22]. The use of DNA biorepositories linked to EMR systems (as in the case of the Electronic Medical Records (EMR) and Genomics Consortium, eMERGE) [82,83] could reduce efforts and costs, facilitating extensive analysis of the whole genome (GWAS) or exome (WES) [22,82,83]. Phenotypic variability can create difficulties in investigating the complex and heterogeneous genetic architecture of LEAD. Two subtypes of LEAD, distal and proximal, are described, associated with a different profile of risk factors and associated comorbidities [22]. Proximal LEAD is more commonly associated with smoking, dyslipidemia, hypertension, and the female sex, while distal LEAD is more commonly associated with older age, male sex, and DM [18,92]. Differentiation of LEAD patients according to location is possible through non-invasive arterial doppler ultrasound and according to the presence or absence of DM [18,92].

#### 5.1.1. Interactions between Genes (Epistasis) and between Genetic and Environmental Factors

Taking into account that the complex pathophysiological mechanism of LEAD includes changes in different atherogenic pathways, it is considered unlikely that the etiology is monogenic, most likely involving several genes with additive effects (polygeny), which interact both with each other (epistasis), as well as with environmental risk factors (smoking, sedentary lifestyle, dyslipidemia, and DM). The identification of the multitude of loci and genes of susceptibility to LEAD and the combination of them, constitutes a challenge for future research. Smoking is considered an environmental factor with a major risk for LEAD, but the different susceptibility to LEAD in the case of smokers suggests that the interaction with genetic factors can influence the evolution and progression of LEAD [18,22,93].

Previous studies have focused on analyzing the association between smoking [93,94] and obesity [94], with susceptibility loci for LEAD [93,94]. In the future, studying the association between DM, ESRD, and LEAD could shed light on the still unknown aspects of the pathophysiological mechanisms of LEAD [22].

To date, there are several GWASs that have analyzed the relationship between genetic and environmental factors in CVD [94,95,96]. Although several statistical methods were proposed for the evaluation of gene−gene and gene−environment interactions, they identified few such interactions, which were later replicated in other studies [18]. In the future, with the advent of GWASs that will analyze more and more samples (hundreds of thousands or even millions of samples), it will be feasible to conduct a powerful genome-wide gene−environment interaction analysis [18,22].

#### 5.1.2. Rare Allelic Variant Association Studies

Common allelic variants of genetic susceptibility do not fully explain the heritability of polygenic, complex diseases, including LEAD. The “missing heritability” could also be explained by the intervention of some rare allelic variants (defined as minor allele frequency <1%), but it is not precisely known how they contribute to the genetic susceptibility for LEAD.

GWASs present a major limitation, represented by the fact that the allelic variants identified could be in a linkage disequilibrium (LD) with the disease-causing alleles. Thus, they do not provide any information about the associated pathophysiological mechanism of the disease, which must be identified later through specific functional studies. Moreover, the definition of the studied clinical phenotype may vary in different study groups. GWASs can only identify high-frequency common alleles (5%) that have small additive effects, which are less efficient at identifying rare genetic variants with a larger effect size [22,97].

GWASs (SNPs) have a low power to identify structural abnormalities (copy number variations (CNVs) caused by translocations, insertions, and deletions), as well as to identify interactions between different genes (epistasis) or between genetic and environmental factors. Exceeding these limits requires different strategies: correctly defining the phenotype for the analyzed cases, increasing the number of analyzed samples and the use of groups with extreme phenotypes; increasing the sensitivity of the detection rate, especially of rare allelic variants, by developing powerful biostatistical tools; taking into account the mechanism of epigenetic regulation of gene expression; fine mapping of SNPs or using next generation sequencing (NGS) for regions of interest to identify rare allelic variants and/or structural variants [18,22,38,46,97].

Association studies of rare variants could be an objective for future research related to genetic susceptibility to LEAD. This approach was successfully illustrated in the study by Huyghe et al. [98]. Using exome array genotyping the authors identified rare allelic variants that determine insulin resistance [98]. The conclusion of the study was that exome array genotyping is a valuable approach to identify low-frequency variants that contribute to complex disease [18,98].

Whole Genome Sequencing (WGS) and Whole Exome Sequencing (WES)

Whole genome (WGS) or exome (WES) sequencing studies represent the next target in terms of the genetic analysis of LEAD [18,22,98]. These have proven useful, especially in cases of LEAD with familial aggregation [99,100]. Future GWASs that will use array data will reveal new risk loci for LEAD through associations of common variants (minor allele frequency >0.01) and extended analyses such as WES and WGS will allow the identification of genetic markers–rare disease-causing variants of larger effect size. The loss-of-function variants are of the greatest interest, allowing a direct correlation between the allelic variant and the gene that determines susceptibility to LEAD. The ability to identify these disease-informative risk alleles will increase as biobanks continue to invest in extensive exome or whole genome sequencing studies [18,22].

In a WES study performed in the case of two siblings with aortic hypoplasia, atherosclerosis, systolic hypertension, and premature cataracts, a rare homozygous mutation (Ser818Cys) was identified in the *INO80D* gene, located on chromosome 2q33.3 [25,99]. It encodes INO80D, a subunit of the human INO80 chromatin remodeling complex, which is involved in the physiology and development of the cardiovascular system [17,22,25]. The authors suggested that there could be a link between the Ser818Cys mutation in the *INO80D* gene and accelerated arterial aging [99].

#### 5.1.3. Epigenetics, Differential Gene Expression, Modifier Genes, Pleiotropy

GWASs have begun to elucidate the LEAD heritability, initially estimated at ≈20% [17]. The deciphering the mechanisms through which genetic and environmental factors interact have extended LEAD heritability to about 20–58% [17,18]. An important aspect that must be considered regarding the LEAD heritability is represented by the mechanisms of epigenetic regulation of gene expression. Epigenetics is the science that explores the control of gene expression without changing the basic DNA sequence through three mechanisms: methylation of CpG islands (through DNA methyltransferases, DNMts), post-translational modifications of histones (PTMs), and microRNAs (miRNAs) [17,18,19,22]. All three epigenetic mechanisms are well documented in vascular biology. DNA methylation changes have been demonstrated in homocysteine-induced atherosclerosis [101], VSMc proliferation and migration which leads to intimal hyperplasia and restenosis [102], flow-dependent atherosclerosis [103], and vascular calcification [17,102,104].

Histone modifications are shown to be involved in vascular homeostasis (regulated by the *KLF2* gene) [105], the interaction between endothelial cells and vessel wall reactive oxygen species [106], vascular remodeling in response to laminar stress flow [107], and vascular endothelial response to hypoxia [17,108].

To date there is evidence indicating that miRNAs are closely involved in the complex pathophysiological mechanisms of atherosclerosis including endothelial dysfunction, angiogenesis and vascular remodeling, lipid accumulation, inflammation, thrombosis, and calcification [17,20,109,110]. Knowing and deciphering the mechanisms by which miRNAs intervene in the formation and progression of the atherosclerotic plaque, creates new hopes related to the diagnosis, prognosis, and treatment of atherosclerotic vascular disease [109].

It is proven that many of the LEAD risk factors intervene and regulate gene expression through epigenetic mechanisms. Increased hemodynamic forces have been shown to be able to induce DNA methylation and histone modification in VSMc [111]. Despite this evidence, the mechanisms by which DNA demethylation or histone modification are involved in the specific regulation of genes involved in the production of hypertension, are not fully elucidated [111,112,113]. Diabetes mellitus (DM) affects DNA miRNA expression and histone acetylation [114]. In DM, the basic epigenetic mechanisms are deregulated in all target organs through chronic inflammation, oxidative stress and the alteration of growth factors. Epigenetic changes in DM play a significant role in delayed wound healing and transgenerational transmission of the disease. Early identification and disabling of mutated epigenetic mechanisms with advanced epigenomic tools could represent the most effective management measure for DM and its complications in the near future [114]. In many studies, smoking has been shown to affect DNA methylation [115,116]. The aging process has been linked to DNA methylation and the regulation of gene expression by miRNAs [110,117].

No studies investigating the direct link between epigenetic changes and LEAD have yet been reported. It seems that the sedentary lifestyle is correlated with an unfavorable epigenome. Exercise has been shown to increase DNA methylation in peripheral blood cells and alter histone acetylation (lysine 36-H3 acetylation) [17,118,119]. Other studies suggest that exercise induces both dynamic changes in H4 deacetylation/loss of promoter methylation and transient changes in miRNA expression profiles [17,118]. Identifying the dynamics of epigenetic regulation could allow for effective therapeutic interventions involving the direct modulation of gene expression by modifying environmental factors [17,119,120]. Masud et al. [121] showed that genes that intervene in different signaling pathways and influence the immune response, the inflammatory mechanism, and apoptosis are differentially expressed in peripheral blood mononuclear cells of LEAD patients [121]. In addition, RNA sequencing allowed for a better understanding of the pathophysiological mechanism of the disease, as well as the response to treatment [18]. Numerous studies have discussed the existence of modifier genes of the LEAD phenotype, these being involved in different mechanisms that intervene in the pathophysiological process of LEAD [18].

The occurrence of PAOD and CLTI correlates with the number of risk genotypes present simultaneously in the same individual. An example of how disease susceptibility results from the interaction between a series of modifier genes present simultaneously in the same individual is represented by the polymorphism of the genes that encode the molecules involved in the inflammatory process (*IL6, SELE, ICAM1, MCP1/CCL2, MMP1, MMP3*) [45]. In this context, the authors concluded that synergistic effects between proinflammatory genotypes could be potential markers for the presence and severity of atherosclerotic disease [45].

The *NOS3* haplotype −786/4a and the *ACE* D allele increase the susceptibility to LEAD in smokers (but not in non-smokers), which is evidence for the interaction between modifier genes and environmental factors [55].

There are studies that attest to the fact that certain genes involved in LEAD etiopathogenesis have pleiotropic effects, causing manifestations in other types of atherosclerotic diseases manifested in other vascular beds [18]. Helgadottir et al. [29] showed that the 9p21 locus correlates with both LEAD and atherosclerotic CAD, abdominal aortic aneurysms (AAA), and intracranial aneurysms [29]. Gretarsdottir et al. [32] showed that the A allele of rs7025486 on 9q33 (*DAB2IP* gene) was associated with AAA, early-onset MI, and LEAD [32].

The strongest pleiotropic locus was *ATXN2/SH2B3*, at the level of which, Li et al. [121] found three SNPs (rs653178, rs4766578 and rs3184504) in near-complete LD (r^2^ = 0.99) associated with a group of CVD, DM, and hematological and autoimmune disorders [18,122]. The allelic variants identified in patients with atherosclerotic disease manifested in other vascular beds could also contribute to the susceptibility to LEAD, forming the basis of future research in order to decipher the genetic architecture of LEAD [18].

#### 5.1.4. Translating the Results of GWASs into Clinical Practice and The Importance of Polygenic Risk Scores (PRS) for Prevention of LEAD

As future GWASs analyze larger cohorts of LEAD patients and common polymorphisms/alleles associated with a susceptibility to LEAD are identified, biostatistical processing of the obtained data will permit the achievement of polygenic risk scores (PRSs) [22]. A PRS is made when multiple genetic variants previously identified as specific for a particular disease are combined into a single tool [22]. PRS is the aggregate contribution of many common genetic variants (minor alleles with a frequency >0.01) that have small to moderate individual effects. PRS are single, quantitative values for genetic susceptibility to a polygenic, complex disease; for example, LEAD PRSs are generated using statistics from GWASs for a given disease and use both the number of risk variants and the size of the associated effect [22].

Two consortia funded by the National Human Genome Research Institute are addressing major knowledge gaps in PRSs. The eMERGE Network [84,123,124], investigates how PRSs for common diseases can be implemented in current clinical practice. The Polygenic RIsk MEthods in Diverse populations (PRIMED) Consortium is working to improve the methods and application of PRSs in different ancestry populations [125]. The group will collectively integrate data on a broad range of phenotypes from over 120 datasets across 45 countries.

The polygenic risk score for LEAD had pleiotropic effects and was associated with a greater OR (odds ratio) of CAD, heart failure, and cerebrovascular disease [126,127].

Wang et al. [126] used statistical data from the largest LEAD GWAS from the Million Veteran Program and performed PRSs with genome data from UK Biobank [126]. The authors evaluated the clinical utility of adding the best-performing PRSs to a LEAD clinical risk score. They demonstrated that adding a LEAD PRS to clinical risk models can help improve the detection of present but undiagnosed disease. Using a genome-wide PRS can discriminate the risk of LEAD and other CVD [126].

Kullo et al. [127] analyzed the data reported by Wang et al. [126], considering that the information presented by them adds LEAD to the list of CVD for which a PRS has been reported, and demonstrates the incremental predictive (but modest) power of the LEAD score [127].

However, an additional effort will be needed to use PRS in clinical practice, including increasing the number of analyzed samples and controls in GWASs of LEAD. Also, the use of new methods of genetic analysis will increase the predictive power in various ancestry groups [127]. Most PRSs are derived from studies featuring populations of European ancestry. It is necessary to develop robust PRSs for various population groups, in order to avoid differences in terms of prevention measures in maintaining the health of the population [27]. In the case of LEAD, PRSs could be used to identify the high-risk asymptomatic individuals for the purpose of establishing effective prophylactic therapy or for people with IC who have an increased risk of developing CLTI and limb amputation, associated with increased mortality [22,127].

#### 5.1.5. Mendelian Randomization

Studies have been reported attesting to the utility of Mendelian randomization (MR) studies for elucidating the relative contribution of different risk factors for atherosclerosis in each vascular bed [22]. Mendelian randomization (MR) studies can identify a causal relationship between a biomarker and LEAD, providing evidence of the contribution of the biomarker to the development of the disease, specifying whether the observed association is influenced by unrecognized exogenous factors or the disease itself affects the biomarker level [22].

The genetic variant used in MR should significantly affect the investigated biomarker, but should not affect other phenotypes, which could confound the association between biomarker and disease. If the biomarker is a true causal risk factor for LEAD, the genotypes of the genetic variant used should be associated with LEAD risk in the direction predicted by the association of the biomarker with LEAD [22]. Analysis of the etiological factors of LEAD by MR has tremendous potential to identify possible therapeutic targets. MR has been widely used in GWASs to identify causal relationships between risk factors (hyperlipemia, hypertension) and CAD or abdominal aortic aneurysm [22].

In the case of LEAD, MR studies have analyzed the contribution to the development of LEAD in the case of factors involved in hemostasis and thrombosis (F5 Leyden) [128]. Small et al. [128] demonstrated, using a two-sample MR, that elevated factor VIII (FVIII) values (OR 1.41) [95% CI, 1.23–1.62; *p* = 6.0 × 10^−7^) and VWF (von Willebrand factor) (OR 1.28) [95% CI, 1.23; *p* = 6.0 × 10^−7^) were correlated with increased risk of thrombosis and LEAD [128]. The results obtained correlated with the already known supportive role of Factor V Leiden in hemostasis and thrombosis in the case of LEAD [22,128].

Dikilitas et al. [129] performed an MR study of a LEAD-related phenotype of polyvascular atherosclerotic disease (defined as atherosclerosis in more than one vascular bed). Although the study did not specifically address LEAD, the authors observed that certain lipid particles, as well as systolic (SBP) and diastolic blood pressure (DBP) values, were causally related to polyvascular involvement in patients with LEAD [129].

Following this study, Levin et al. [130] did an MR study in which they investigated the causal relationship between circulating lipoproteins and the risk for LEAD [130]. The results obtained by them highlighted the fact that, although ApoB remains a probable cause for the pathogenesis of atherosclerosis in LEAD, distinct subfractions of the lipoprotein ApoB (very low-density lipoprotein, VLDL) probably play a differential role in the pathogenesis of LEAD and CAD [130].

Levin et al. [131] analyzed the contribution of smoking in atherosclerotic diseases, including CAD, LEAD, and LAS [131]. They used data from a GWAS of smoking (UK Biobank) [132], GWAS of CAD (CARDIoGRAMplusC4d) [133], GWAS of LEAD (VA Million Veteran Program) [131], and GWAS for LAS (MEGASTROKE) [59]. The authors found that genetic liability to smoking was associated with an increased risk of all three atherosclerotic diseases [131]. The study revealed that the association between smoking and disease risk was significantly higher in the case of LEAD compared to LAS or CAD [131].

In another study, Levin et al. [134] analyzed the effects of systolic (SBP) and diastolic (DBP) blood pressure on the risk for LEAD. The authors showed that the effects of SBP in particular, but also of DBP, are greater in the case of CAD compared to LEAD [134]. Cumulatively, the results of these studies highlight the usefulness of MR for elucidating the relative contribution of atherosclerosis risk factors at the level of different vascular beds [131,132,133,134].

#### 5.1.6. Microbiome

There are studies showing that periodontal disease is associated with atherosclerosis, suggesting that bacteria in the oral cavity may contribute to the development of atherosclerosis and cardiovascular dysfunction [18]. Koren et al. [135] identified by polymerase chain reaction (PCR) the presence of bacterial DNA in atherosclerotic plaques [135]. Wang et al. [136] suggested an association between gut microbiome-derived metabolites (choline, betaine, and trimethylamine-N-oxide) and the presence of CAD [136]. These observations could be the basis of future studies in patients with LEAD, in order to identify new mechanisms, some metabolomic markers of the initiation and progression of the atherosclerotic disease, as well as the identification of new therapeutic targets, which involve the modification of the intestinal or dental flora by changing the diet and the administration of probiotics [18,136,137].

#### 5.1.7. Prophylactic Measures for Patients and Families at High Risk for LEAD

Traditional risk factors for LEAD are smoking, diabetes, high BP, and high levels of LDL-C (low-density lipoprotein-cholesterol) and obesity [17,18,19,22]. There is evidence that the magnitude of the association of smoking with the onset of the disease is greater for LEAD compared to CAD, and passive smoking is an independent risk factor for LEAD. In addition, there has been a gradual association between the number of modifiable risk factors and the prevalence of LEAD and a long-term association between initial risk profiles and LEAD risk after nearly 40 years of follow-up [138]

Primary prevention in the case of LEAD addresses the modifiable risk factors of LEAD and includes avoiding smoking (active and passive), lipid consumption (low-fat diet of the Mediterranean type), physical exercises, avoiding obesity, and the correct treatment of DM [138].

Compared to CAD, there are fewer studies on the effectiveness of preventive measures and therapy used in the secondary prevention of LEAD. However, since atherosclerosis is a multisystemic disease with similar etiopathogenesis in different vascular beds, it is expected that the treatment of risk factors will have systemic effects, including in the case of LEAD [138].

ABI measurement is recommended as a first-line noninvasive test for the screening and diagnosis of LEAD. An ABI < 0.90 is an independent predictor of cardiovascular events and is useful to identify patients at a moderate to high risk of CVD. However, there is insufficient evidence to assess the benefits of screening for LEAD with ABI in asymptomatic adults. Lifestyle changes, including quitting smoking, dietary changes (low-fat diet), physical activity, optimizing diabetic control, and correcting hyperlipidemia and hypertension are the basis of secondary prevention in LEAD patients [138,139,140].

Reverse associations with LEAD have been reported for some subtypes of dietary fats, fiber, antioxidants (vitamin E and C), folic acid, vitamins B6, B12, and D, flavonoids, and fruits and vegetables [138]. There are studies attesting that the low-lipid Mediterranean diet, could be effective in the primary and secondary prevention of LEAD, but further studies are needed to clarify this association [138].

Identification of some biochemical markers (low serum levels of 25-hydroxyvitamin D, hyperhomocysteinemia, low levels of serum bilirubin, total adiponectin, and lipoprotein-associated phospholipase A2) is also an important aspect of LEAD prevention, attesting to a proatherosclerotic risk profile characterized by endothelial and systemic dysfunction in LEAD patients. Several markers of inflammation, such as serum C-reactive protein (CRP) and IL6 (interleukin-6) are associated with symptomatic LEAD independently of traditional cardiovascular risk factors [43,138,141].

In the study of the National Health and Nutrition Examination Survey (NHANES) 1999–2002, which included the general adult population of the United States, Wildman et al. [142] demonstrated the existence of a gradual relationship between the analyzed inflammatory markers (C-reactive protein, fibrinogen, and leukocyte count) and LEAD [142].

Studies done to date have indicated that prophylactic management of LEAD patients with antiplatelet and antithrombotic drugs, statins, and angiotensin-converting enzyme (ACE) inhibitors prevents local disease progression, reduces cardiovascular events, and improves prognosis, as well as in the case of patients with atherosclerotic CAD [139,143].

Consequently, the therapeutic measures used in the secondary prevention of atherosclerosis in patients with LEAD are as important as those used in patients with coronary or cerebrovascular disease, improving prognosis and life expectancy [144].

LEAD treatment aims to reduce symptoms in patients with IC and prevent progression to CLTI and limb amputation [137,143]. Most of the time, lifestyle changes, physical exercises, and treatment for IC are enough to slow the progression or even reverse the symptoms of LEAD [145].

There are studies that address gene therapy used to improve blood flow by restoring blood vessels in affected limbs. The effectiveness of this treatment in reducing the risk of amputation or improving the quality of life is not fully known; further studies are needed in the future [146].

#### 5.1.8. Genetic Counseling in Peripheral Artery Disease Patients

Recurrence risk assessment in cases with familial aggregation or in the descendants of an individual with LEAD, as well as genetic counseling can be difficult, due to the complex, multifactorial etiology of LEAD, especially since the genetic component is represented by a large number of loci and genes of susceptibility. Moreover, added to this is the variable interaction between different genes (epistasis), as well as the interaction between genetic and environmental factors. Genetic counseling for patients at a high risk of LEAD should include, in addition to a detailed physical examination, family history and a genealogical analysis (which can provide important information about family aggregation), personal medical history, habits, and medications used. There are currently vascular medicine centers that offer genetic testing as a diagnostic tool in their approach to treating patients with vascular conditions, including LEAD.

Although our study focused on the in-depth analysis of data from the specialized literature regarding the role of LEAD genetic risk factors, as well as on their analysis methods, we believe that the main limitation of this study was the insufficient data in the literature related to the etiology of LEAD (the “missing heritability” of LEAD).

This could be due, on the one hand, to the large number of genes and susceptibility loci for LEAD, and on the other hand, to the insufficient exploration of gene−gene interaction and interactions between genetic factors and environmental factors.

In addition, the genetic bases of non-atherosclerotic LEAD (PAOD), such as autoimmune vasculitis and fibromuscular dysplasia which have pathophysiological mechanisms similar to LEAD (inflammation, vascular remodeling, VSMc proliferation), were not discussed. The study of these mechanisms could also shed light on atherosclerotic LEAD [18,22].

Deciphering the etiology of LEAD is a fascinating, current topic that remains relevant due to the complexity of the possible factors involved and their interaction. In addition, future research will most likely identify new candidate LEAD biomarkers, which, together with the use of PRS, will improve LEAD risk prediction. In addition, it is not known to what extent the data obtained from small studies which have not been replicated in other studies are reflected in the general population or if there are population differences depending on ethnicity.

In the case of LEAD, we cannot yet speak of a personalized medicine approach, and we do not know to what extent the detected genetic baggage is reflected in the therapeutic response. Starting from this aspect, in the future, modern medicine will pave the way for the implementation of genetic investigations in the primary prevention of CVD, including LEAD. As a future strategy, stem/progenitor cell transplantation could represent a potential therapy to induce angiogenesis in ischemic tissue, thus preventing severe complications and limb amputation in patients with advanced LEAD [147].

## 6. Conclusions

In recent years, important progress has been made in the elucidation of the complex etiology of LEAD with the help of increasingly advanced molecular technologies and the processing of data obtained through efficient statistical and bioinformatics methods. Although the overall mortality of LEAD remained high, the main benefit was the identification of new therapeutic targets and the improvement of prognosis and life expectancy.

GWASs generated important information that provided a new perspective on the pathophysiological mechanisms of LEAD and created the premises for deciphering the complex genetic architecture of the disease. Currently, existing data show that newly identified genetic factors, together with those already known, account for approximately 20–58% of the heritability of LEAD.

In the future, the identification of new genetic factors could explain the “missing heritability” of LEAD, without ignoring the fact that the interaction between different genes (epistasis) or between genetic and environmental factors as well as the existence of modifier genes could contribute to the phenotypic variability of LEAD. An important objective of future research could be the identification of genetic factors that would have a protective role against LEAD, as well as studies on the effectiveness of gene therapy in order to restore the circulation of the affected limbs.

It is expected that following the sustained efforts of large international consortia, new genetic risk factors for LEAD will be identified, with translation into clinical practice related to the development of new therapeutic molecules that act on specific targets.

The creation of large and accessible international databases and the sharing of information between different study groups will elucidate the conflicting results of some studies that included small groups of patients. Elucidation of ethnic differences in LEAD risk and response to various therapies may indicate genetic differences that will allow for the development of targeted and personalized drug therapy.

As the amount of data provided by different types of studies increases, improved statistical and bioinformatics analysis methods will be needed to help decipher the complex etiology of LEAD.

In the future, the widespread use of PRSs could improve the prediction of LEAD risk, allowing for the identification of individuals at a higher risk of LEAD who could benefit from preventive measures and personalized treatment.

## Figures and Tables

**Figure 1 ijms-23-10481-f001:**
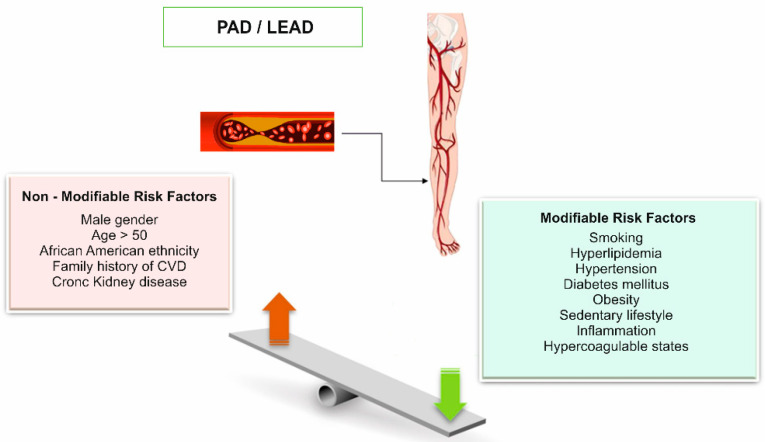
Risk Factors for Lower Extremity Peripheral Arterial Disease [1,2,3,4,6,7,8,12,14,15] CVD: Cardiovascular disease.

**Figure 2 ijms-23-10481-f002:**
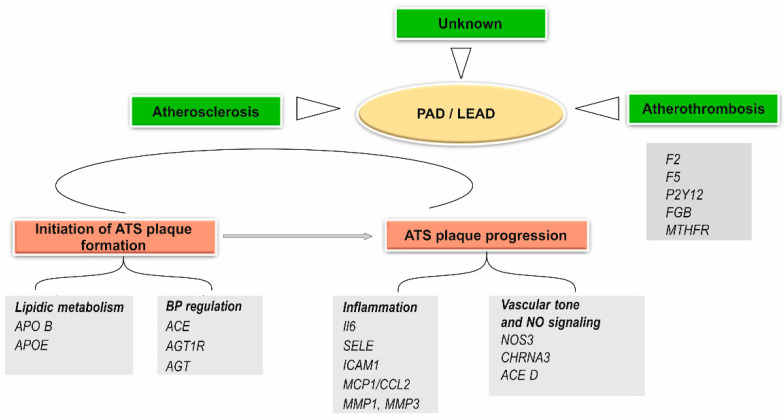
The Main Genes Involved in the Pathophysiological Mechanism of LEAD [13,17,22]. ATS plaque: atherosclerotic plaque; BP: blood pressure.

**Table 1 ijms-23-10481-t001:** Genetic Etiology of Lower Extremity Peripheral Artery Disease.

Polymorphism	Type of Study	Diagnosis	Location/ Chromosome	The Nearest Gene (s)	References
*LA studies/SNPs*					
	LA/SNPs	LEAD/POAD	1p31	Unknown	[23]
	LA/SNPs	LEAD	1p	*HTR6 5, PLA2G2E, ECE1, IL22RA1, ECE1, IL28RA, LDLRAP1, PTAFR, FABP3, COL16A1, FNDC5, COL8A*	[24]
	LA/SNPs	LEAD	3p,3q	*IL1RAP, FGF12*	[24,25]
	LA/SNPs	LEAD	6q	*LPA, PLG*	[24]
	LA/SNPs	LEAD	7q	*Unidentified*	[24]
	LA/SNPs	LEAD	10p	*IL15RA, ITGA8*	[24]
	LA/SNPs	LEAD	16q	*IL4R, IL21R*	[24]
*CGS/SNPs studies*					
rs2171209	CGS/SNPs	LEAD/ABI	6q25.3	*SYTL3*	[25,26]
rs290481	CGS/SNPs	LEAD/ABI	10q25.2–q25.3	*TCF7L2*	[25,26]
rs3745274	CGS/SNPs	LEAD/ABI	19q13.2	*CYP2B6*	[24,25]
rs891512, rs1808593	CG/SNPs	LEAD	7q36.1	*NOS3*	[25,27]
rs1042713	CGS/SNPs	LEAD/Lp(a)	5q32	*ADRB2*	[24,25,27]
rs828853, rs1299142	CGS/SNPs	LEAD/DM	2p13.1	SLC4A5	[25,27]
rs3917187/rs2284791, rs668871	CGS/SNPs	LEAD/TG	14q24.3, 6q25.3	*TGFB3*, *SLC22A3*	[24,25,27]
rs2110981, rs2270042	CGS/SNPs	LEAD CRP	2p13.3	*ADD2*	[25,27]
rs22394, rs10247, rs11681	CGS/SNPs	LEAD/FB	2p13.3	*ATP6B1*	[25,27]
rs207129, rs154027, rs257376	CGS/SNPs	LEAD/Hyc	6p22.2, 7q22.3	*SLC17A2*, *PKRAR2B*	[24,25,27]
*Relevant LEAD-related GWAS studies*					
rs10757269	GWAS^1^	CAD LEAD/ABI	9p21	*CDKN2B*	[25,28,29,30]
rs13290547	GWAS^1^	LEAD	9q33	*DAB21P*	[25,28]
rs3794624	GWAS^3^	LEAD	16q24.2	*CYBA*	[25,28]
rs1122608	GWAS^3^	LEAD	19p13.2	*LDLR*	[24,25,28]
rs10757278	GWAS^1^	AAA/CAD/ LEAD/ICA	9p21	*CDKN2A/CDKN2B*	[29]
rs1333049	GWAS^1^	LEAD/MI	9p21	*CDKN2A/CDKN2B*	[28,30]
rs1051730	GWAS^3^	No. of CS	15q25.1	*CHRNA5/A3/B4*	[25,31]
rs7025486	GWAS^3^	AAA/LEAD/MI	9q33	*DAB2IP*	[25,28,32]
rs3794624	GWAS^1^	LEAD	16q24.2	*CYBA*	[25,28]
rs376511	GWAS^3^	TAO	3p25.3	*IL17RC*	[25,33,34]
rs7632505	GWAS^3^	TAO	3q21.1	*SEMA5B*	[25,33,35]
rs10178082	GWAS^3^	TAO	7p21.3	*RPA3*	[25,33]
rs1902341	GWAS^1^	LEAD	3p23–p22.3	*OSBPL10*	[25,36]
rs9584669	GWAS^1^	LEAD	13q32.2	*IPO5/RAP2A*	[17,25,37,38,39]
rs6842241	GWAS^1^	LEAD	4q31.22–q31.23	*EDNRA*	[25,37]
rs2074633	GWAS^2^	LEAD	7p21.1	*HDAC9*	[25,37]
rs653178/rs3184504	EMR GWAS^2^	LEAD/MI	12q24.12	*ATXN2-SH2B3*	[25,40]
rs2554503	GWAS^3^	LEAD	8p23	*CSMD1*	[25,36]
rs235243	GWAS^3^	LEAD	1p36.22–p36.21	*VSP13D*	[25,36]
*Other Genetic Polymorphisms Associated with the Pathophysiological Mechanisms of LEAD*					
Genes involved in lipid metabolism		LEAD	2p24.1 19q13.32	*APOB* *APOE*	[25,41] [25,42]
Genes involved in the mechanism of inflammation		LEAD	7p15.3 1q24.2 19p13.2 17q12 11q22.2	*IL6* *SELE* *ICAM1* *MCP1/CCL2* *MMP1, MP3*	[13,43,44] [13,45] [25] [13,25,45]
Genes involved in blood pressure regulation		LEAD	17q23.3 3q23 1q42.2	*ACE* *AGT1R* *AGT*	[13,25,46,47,48,49,50]
Genes involved in the function of VSMc		LEAD	7q36.1	*NOS3*	[25,51,52,53,54,55]
Genes involved in vascular homeostasis		LEAD/DM/AT	20q13.12	*SLC2A10*	[13,25,56]
Genes involved in atherothrombosis		LEAD	11p11.2 1q24.2 3q21.5 4q31.3	*F2* *F5* *P2YR12* *FGB*	[25,57,58] [25,58,59] [25,60,61] [25,62,63]
HHcy		LEAD	1q36.22	*MTHFR*	[25,64,65]

LA: Linkage analysis; CGS: Candidate gene-based study; GWAS: Genome-wide association study; GWAS^1:^ GWAS significant (*p* < 5 × 10 ^−8^) loci; GWAS^2^: GWAS loci of board line significance (*p* < 1 × 10 ^−7^); GWAS^3:^ GWAS additional loci of potential interest; SNP: Single nucleotide polymorphism; AAA: Abdominal aortic aneurysm; ABI: Ankle-brachial index; EMR: Electronic medical record; POAD: Peripheral arterial occlusive disease; DM: Diabetes mellitus; TG: triglycerides; FB: fibrinogen; CRP: C-reactive protein; Lp(a): Lipoprotein (a); ICA: Intracranial aneurysms; MI: Myocardial infarction; PE: Pulmonary embolism; VSMc: Vascular smooth muscle cells; HHcy: Hyperhomocysteinemia; AT: Arterial tortuosity syndrome; TAO: thromboangiitis obliterans; CS: cigarettes smoked.

**Table 2 ijms-23-10481-t002:** Classification of Clinical Forms of Acute and Chronic Limb Ischemia [1,2,66,67].

Clinical Features	Fountain Classification	Rutherford Classification	
	Stage	Grade	Category
Asymptomatic	I	0	0
Mild claudication	IIa	I	1
Moderate claudication Severe claudication	IIb	I	2
		I	3
Ischemic rest pain	III	II	4
Minor tissue loss (Ischemic ulcers of the digits of the foot)	IV	III	5
Major tissue loss (severe ischemic ulcers or gangrene)		IV	

## Data Availability

Not applicable.

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
