# Peer review of "The Genetic Architecture of the Etiology of Lower Extremity Peripheral Artery Disease: Current Knowledge and Future Challenges in the Era of Genomic Medicine"

_ijms, 2022, doi:10.3390/ijms231810481_

Round 1

Reviewer 1 Report

I have read the review with a great interest. Genetic aspects of PAD are very  interesting. The manuscript provides important contribution in this area.

The paper is well arranged, but in my opinion too long. I would suggest to delate some paragraphs that are not relevant to the review issue, such as classification of PAD severity. 

Also, I think that a Table should be added on the genes and the role/ action of the respective genes in PAD etiology, rather, instead of very long section 4.2.4a.b.c.d etc.

Author Response

Dear Reviewer, thank you for your time evaluating our work and for the point raised.

  1. Given that we just briefly discussed the two Rutherford and Fontain classes in the text, we believe the table that lists the PAD/LEAD severity classification to be relevant. The table does not duplicate the material in this manner. We should explain in the text what the two classes mean without referring to the appropriate table. Additionally, it is recognized that not all IC patients progress towards CLTI, a condition that may be related to underlying genetic factors.
  2. We consider that table 1 summarizes the genetic factors (genes) involved in the etiology of PAD/ LEAD, correlated with the physiopathological mechanism in which they intervene (inflammation, coagulation, formation of atherosclerotic plaques, blood pressure regulation, and vascular homeostasis), as well as the types of studies through which were identified (LA, GWAS; WES). In Section 4, we reviewed some of the relevant studies that have been published in the literature regarding the contribution of these genetic factors to the pathophysiology of PAD/LEAD, emphasizing the importance of the interaction between different genes (epistasis), of the interaction of genes - environmental factors, as well as the role epigenetic regulation of gene transcription.

Thank you!

Reviewer 2 Report

The authors present a review on LEAD etiology regarding genetic regulation. 

The review is thematically well prepared, however, some major points have to be addressed:

 Please be consistent with the nomenclature: PAD is not only LEAD,  as you correctly cite in the introduction. However, please adapt the sentence in line 51: Peripheral artery disease (PAD) also called Lower extremity artery disease (LEAD) --> e.g. Lower extremity artery disease as part of PAD

please consider this throughout the text.

Additional remarks: Could you comment on the genetic regulation and endothelial (dys-)function?

Author Response

Dear Reviewer, thank you for your time evaluating our work and for the point raised.

  1. As a result of your suggestion, we reformulated the opening sentence (line 51) and replaced the term PAD with LEAD throughout the text.
  2. Regarding the genetic factors that are involved in endothelial dysfunction, we have already discussed the polymorphism of the NOS3, SLC2A10 genes (involved in vascular homeostasis), including the role of miRNAs and hyperhomocystinemia.

Thank you!

Reviewer 3 Report

This extensive review on genetic architecture in PAD is of great interest. Although far from competent to evaluate the genetic studies that are cited, my understanding is that the topic is very well covered. 

The introduction describes and cites clinical facts of PAD in an appropriate way, however, I find a comment with references 6 and 13 less appropriate: 

"However, there is no consensus for the use of screening even in patients at high risk for 228 PAD (smokers or diabetics), because there is a lack of clinical benefits regarding the treatment of PAD [6,13]."

I agree screening is probably not reasonable, due to the fact that asymptomatic PAD does not benefit from antiplatelet treatment.  This is, however, not clear from your text

Author Response

Dear Reviewer, thank you for your time evaluating our work and for the point raised.

At your suggestion, we reformulated the phrase / paragraph regarding the utility of PAD / LEAD screening in the case of asymptomatic patients, according to the information available in the AHA/ACCC Guideline 2016 [6].

 Thank you!

Reviewer 4 Report

Butnariu and colleagues reviewed many studies, which tried to identify a genetic basis for Peripheral artery disease (PAD). The manuscript is overall well-written. Some minor errors should be corrected:

Line 127: typos and error in grammar

Paragraph line 143ff: This is unclear and does not seem practical.

Line 161: depends on the skills of the surgeon, but in general not all amputations are lethal.

Line 249: not all regulators of gene expression are epigenetic factors, most are transcription factors.

Line 489: TAO and PAD are pathophysiological different

Conclusion: please specify what you have in mind for the future regarding “preventive measures and personalized treatment

Author Response

Dear Reviewer, thank you for your time evaluating our work and for the points raised.

  1. According to the literature, identifying PAD/LEAD risk alleles and using PRSs to identify those who are at risk for PAD/LEAD, similar to how CAD risk alleles can be used to identify people, could be an objective of future research with the aim of early diagnosis and the implementation of effective prophylactic measures by avoiding PAD-promoting environmental factors. [22, 126,127].

Bibliography

  1. Klarin, D.; Tsao, P.S.; Damrauer, S.M. Genetic Determinants of Peripheral Artery Disease. Circ Res. 2021,128(12),1805-1817. doi: 10.1161/CIRCRESAHA.121.318327.
  2. Wang, F.; Ghanzouri, I.; Leeper, N.J.; Tsao, P.S.; Ross, E.G. Development of a polygenic risk score to improve detection of peripheral artery disease. Vasc Med. 2022, 27(3), 219-227. doi: 10.1177/1358863X211067564.
  3. Kullo, I.J. Polygenic risk score for peripheral artery disease: A tool to refine risk stratification. Vasc Med. 2022, 27(3), 228-229. doi: 10.1177/1358863X221080191.

2. Regarding the secondary prophylaxis of PAD/LEAD, it is already mentioned in the text that it could include ABI screening for asymptomatic patients, low-lipid diet, lifestyle modification, diabetes control, smoking cessation. Also, the use of biochemical markers (inflammatory markers; markers indicating the atherosclerotic profile) for the early diagnosis of PAD have been mentioned. The use of antiplatelet and antithrombotic drugs, statins, and angiotensin-converting enzyme (ACE) inhibitors prevents local disease progression, reduces cardiovascular events, and improves prognosis. In the future, the identification of new genetic factors could lead to the development of new, innovative therapies that act in a targeted manner. Knowing the mechanism of the disease and the specific genetic defect of the patient, the treatment of the patients could be personalized.

3. At your suggestion, we indicated that personalized therapy in the future might involve stem/progenitor cell transplantation as a viable treatment to stimulate angiogenesis in ischemic tissue, preventing serious consequences and limb amputation in patients with advanced PAD/LEAD.

Thank you.